# Id2 Represses Aldosterone-Stimulated Cardiac T-Type Calcium Channels Expression

**DOI:** 10.3390/ijms22073561

**Published:** 2021-03-30

**Authors:** Jumpei Ito, Tomomi Minemura, Sébastien Wälchli, Tomoaki Niimi, Yoshitaka Fujihara, Shun’ichi Kuroda, Koichi Takimoto, Andrés D. Maturana

**Affiliations:** 1Laboratory of Animal Cell Physiology, Graduate School of Bioagricultural Sciences, Nagoya University, Aichi 464-8601, Japan; pha066@osaka-med.ac.jp (J.I.); mine@fbs.osaka-u.ac.jp (T.M.); tniimi@agr.nagoya-u.ac.jp (T.N.); 2Translational Research Unit, Section for Cellular Therapy, Oslo University Hospital, 0379 Oslo, Norway; sebastw@rr-research.no; 3Department of Experimental Genome Research, Research Institute for Microbial Diseases, Osaka 565-0871, Japan; fujihara@biken.osaka-u.ac.jp; 4Institute for Scientific and Industrial Researches, Osaka University, Osaka 567-0047, Japan; skuroda@sanken.osaka-u.ac.jp; 5Department of Bioengineering, Nagaoka University of Technology, Nagaoka 940-2188, Japan; koichi@vos.nagaokaut.ac.jp

**Keywords:** aldosterone, cardiomyocytes, T-type calcium channels (List three to ten pertinent keywords specific to the article yet reasonably common within the subject discipline.)

## Abstract

Aldosterone excess is a cardiovascular risk factor. Aldosterone can directly stimulate an electrical remodeling of cardiomyocytes leading to cardiac arrhythmia and hypertrophy. L-type and T-type voltage-gated calcium (Ca^2+^) channels expression are increased by aldosterone in cardiomyocytes. To further understand the regulation of these channels expression, we studied the role of a transcriptional repressor, the inhibitor of differentiation/DNA binding protein 2 (Id2). We found that aldosterone inhibited the expression of Id2 in neonatal rat cardiomyocytes and in the heart of adult mice. When Id2 was overexpressed in cardiomyocytes, we observed a reduction in the spontaneous action potentials rate and an arrest in aldosterone-stimulated rate increase. Accordingly, Id2 siRNA knockdown increased this rate. We also observed that CaV1.2 (L-type Ca^2+^ channel) or CaV3.1, and CaV3.2 (T-type Ca^2+^ channels) mRNA expression levels and Ca^2+^ currents were affected by Id2 presence. These observations were further corroborated in a heart specific Id2- transgenic mice. Taken together, our results suggest that Id2 functions as a transcriptional repressor for L- and T-type Ca^2+^ channels, particularly CaV3.1, in cardiomyocytes and its expression is controlled by aldosterone. We propose that Id2 might contributes to a protective mechanism in cardiomyocytes preventing the presence of channels associated with a pathological state.

## 1. Introduction

Aldosterone is associated with the development of heart pathologies, in particular hypertrophy, arrhythmia, and heart failure. Patients with aldosterone excess pathologies such as primary aldosteronism or Cushing’s syndrome are also subject to heart diseases [1,2,3,4,5]. In vivo and in vitro studies have shown that aldosterone directly stimulates an electrical remodeling, altering the expression of cardiac ion channels promoting the development of cardiac hypertrophy and heart failure [6,7,8,9]. Particularly, the expression and activity of both cardiac L-type voltage-gated calcium (Ca^2+^) channel (CaV1.2) and T-type voltage-gated Ca^2+^ channels (CaV3.1 and CaV3.2) are up-regulated by aldosterone in isolated neonatal and adult rat ventricular cardiomyocytes [10,11,12]. In the heart ventricle, the calcium influx through the L-type Ca^2+^ channels sustains the depolarization phase of the cardiac action potential and the excitation-contraction coupling [6]. T-type voltage-gated Ca^2+^ channels drive the pacemaker depolarization of the sinoatrial node of the heart [13]. The re-expression of T-type Ca^2+^ channels in the adult ventricle is associated with the development heart hypertrophy and failure [14].

The regulation of the CaV3.1 T-type Ca^2+^ channels expression by corticosteroid involves in five Glucocorticoid Response Elements (GRE) in the promoter of *cacna1g* gene, encoding for the CaV3.1 T-type Ca^2+^ channels [12,15]. Interestingly, these GRE elements act either as repressor or activator on CaV3.1 expression [15]. The expression of CaV3.1 and CaV3.2 are also regulated by miRNA204 whose expression is upregulated by aldosterone stimulation [16]. miRNA204 downregulates the expression of a transcriptional repressor REST/NRSF that blocks the expression of CaV3.2 T-type Ca^2+^ channels [16].

Inhibitor of differentiation/DNA binding protein 2 (Id2) is another transcriptional repressor that belongs to the family of Helix-Loop-Helix (HLH) transcription regulators [17]. Id2 lacks DNA binding domain, and acts as a dominant-negative factor upon dimerization with bHLH transcription factors [17]. Id2 is important for heart development during embryogenesis [18,19]. Id2 cooperates with Nkx2.5 and GATA4 transcription factors for the development of the cardiac conduction system [20]. In the murine embryonic carcinoma cell line P19, Id2 interacts with and blocks the activity of Nkx2.5 and GATA4, preventing these cells to differentiate into beating cardiomyocytes [21,22].

Since, at the molecular level, aldosterone alters the expression of genes involved in cardiomyocytes remodeling [23], we here studied Id2, an important regulator of heart development conduction system during embryogenesis [18,20,24], as a candidate transcriptional regulator which could affect the expression of genes encoding the L-type and T-type voltage-gated Ca^2+^ channels. We here demonstrate that there is a link between aldosterone stimulation and Id2 regulation. By using different molecular tools, we show that Id2 can function as transcriptional repressor of voltage-gated Ca^2+^ channels and could possibly contribute to a protective mechanism in cardiomyocytes exposed to excess of aldosterone.

## 2. Results

### 2.1. Aldosterone Represses Id2 Expression in Neonatal Rat Cardiomyocytes

We first examined the expression of Id2 in the heart (Appendix A) and observed that Id2 mRNA and protein were found in the atria, septum, and left and right ventricle (Appendix A). The mRNA expression of Id2 increased after birth in the whole heart (Appendix A). We next examined whether aldosterone stimulation could affect the expression of Id2 in neonatal rat ventricular cardiomyocytes. The mRNA level of Id2 was reduced after 24 h upon aldosterone stimulation (Figure 1A). The reduction reach significance from 100 nmol/L or higher doses of aldosterone (Figure 1A). Spironolactone, a mineralocorticoid receptor (MR) antagonist, prevented the effect of aldosterone to Id2 mRNA expression in cardiomyocytes (Figure 1B). Accordingly, aldosterone stimulation (1 µmol/L) significantly reduced the protein amount of Id2 compared to control (Figure 1C). These results suggest that aldosterone trough the activation of MR represses the expression of Id2 in neonatal ventricular rat cardiomyocytes.

### 2.2. Overexpression of Id2 Prevents the Aldosterone-Stimulated Effects on Cardiomyocytes

Aldosterone stimulates an increased spontaneous beating rate on neonatal cardiomyocytes and positively or negatively alters the expression of cardiac ion channels [11,25,26,27]. As Id2 expression is repressed by aldosterone (Figure 1), we next assessed the effects of Id2 overexpression (Figure 1A) or siRNA knock-down (Figure 2B) on the rate of spontaneous action potential in neonatal rat cardiomyocytes (Figure 2C and Appendix A). Stimulation with aldosterone (1 μmol/L) for 24 h resulted in a significantly increased rate of spontaneous action potential compared to control. Id2 overexpression reduced the basal rate of spontaneous beating frequency. When Id2-overexpressing cardiomyocytes were stimulated with aldosterone, the rate of spontaneous action potential was similar to control. On the contrary, in cardiomyocytes treated with Id2 siRNA the rate was significantly higher than control cardiomyocytes. A Luc1 targeting siRNA used as control had no effect on the rate. These results suggested us a possible role of Id2 in the regulation of cardiac ion channels expression. We, therefore, examined the mRNA expression of cardiac ion channels involved in the generation of the spontaneous action potentials (Figure 3 and Appendix A). As previously shown [11,12,26], 24 h aldosterone stimulation resulted in a significant increased expression of CaV1.2, CaV3.1, and CaV3.2 mRNA, encoding for the L- and T-type voltage-gated Ca^2+^ channels. Interestingly, the aldosterone stimulated an increase in CaV1.2, CaV3.1, CaV3.2 was suppressed in Id2 overexpressing cardiomyocytes (Figure 3A).

However, the expression of the *Hcn2* and *Hcn4* mRNAs, encoding the Hyperpolarization-activated cyclic nucleotide-gated channels 2 and 4, and *Kcnh2* and *Kcnd3* mRNA, encoding the hERG and Kv4.3 potassium channels, were not changed by aldosterone stimulation (Appendix A). The voltage-gated sodium channel encoding *scn4a* mRNA, expression was slightly increased without reaching significance (Appendix A). Id2 overexpression increased the expression of *Hcn2* and *Kcnh2*. On the other hand, *Scn5a* and *Kcnd3* expression were reduced upon Id2 overexpression. The siRNA knockdown of Id2 resulted in an increase in CaV1.2, CaV3.1, *Hcn4*, *Scn5a* mRNAs (Figure 3B and Appendix A). However, only the increase in CaV3.1 mRNA by Id2 siRNA reached statistical significance. *Hcn2* mRNA was on the contrary reduced.

These results thus suggest that Id2 plays a role in the expression of cardiac ion channels, particularly the CaV3.1 T-type Ca^2+^ channels in neonatal rate ventricular cardiomyocytes. In addition, the suppression of aldosterone-stimulated increased expression of CaV1.2, CaV3.1, CaV3.2 suggests that Id2 could have a cardioprotective potential upon excess of aldosterone condition.

### 2.3. Id2 Expression Modulate the Voltage-Gated Calcium Currents in Cardiomyocytes

We next tested the effect of Id2 overexpression and siRNA knockdown on the voltage-gated Ca^2+^ channels activity of neonatal rat cardiomyocytes (Figure 4). Aldosterone stimulation with a concentration of 1μmol/L significantly increased the voltage-gated Ca^2+^ currents from −30 to +20 mV, a range voltage in which both T- and L-type Ca^2+^ channels are activated. Id2 overexpression reduced the basal, as well as the aldosterone-stimulated increase in the voltage Ca^2+^ currents (Figure 4A). Inversely, Id2 siRNA knockdown significantly increased Ca^2+^ currents from −30 to −10 mV, a range of voltages in which T-types Ca^2+^ channels are activated (Figure 4B). These results reinforce the hypothesis that Id2 acts as a transcriptional repressor for voltage-gated Ca^2+^ channels expression.

### 2.4. Id2 Represses CaV3.1 Expression at the Transcriptional Level

Among all the tested ion channels expression, CaV3.1 T-type Ca^2+^ channel, was the only whose expression was both significantly reduced by Id2 overexpression, significantly increase upon Id2 siRNA knockdown and whose increased expression by aldosterone stimulation was prevented by Id2. These suggested us that Id2 transcriptional repression may be specifically target CaV3.1. To further confirm that Id2 acts on the regulation of the CaV3.1 gene transcriptional activation, we tested whether Id2 prevents the aldosterone-induced transcriptional activity on *cacna1g* gene, encoding for CaV3.1 T-type Ca^2+^ channel, using a luciferase reporter assay under the control of *cacna1g* 0.798 kb promoter (Figure 5). In neonatal rat cardiomyocytes, aldosterone (1 μmol/L) significantly induced transcriptional activation of *cacna1g* promoter compare to control. Id2 overexpression significantly reduced the basal and the aldosterone-stimulated transcriptional activation. Id2 siRNA knockdown significantly stimulated the increase in *cacna1g* promoter activity. These results suggest that Id2 functions as a transcriptional repressor of CaV3.1 T-type Ca^2+^ channel. 

### 2.5. Id2 Prevents the Aldosterone-Stimulated Increase in Voltage-Gated Calcium Channels in Adult Mouse Heart

To confirm the repressive effect of Id2 on Ca^2+^ channel expression in vivo, we generated a cardiomyocyte-specific Id2 overexpressing mice (Figure 6 and Appendix A). These mice were viable and did not show any apparent phenotype difference with non-transgenic mice littermate (WT) at basal line (Table 1). To evaluate Id2 role in adult heart, osmotic pumps containing aldosterone (60 mg/kg/day) or a saline solution were implanted subcutaneously for 7 days in WT and Id2 transgenic mice. The heart of WT mice treated with aldosterone tended to be heavier but this difference did not reach significance compared to other groups in this period (Table 2, Table 3). In contrast to WT mice, Id2-transgenic mice showed no difference between aldosterone and saline treated mice for CaV1.2, CaV3.1, and CaV3.2 expression. These results corroborated in vivo that Id2 also represses the aldosterone-stimulated increase in L-type and T-type voltage-gated Ca^2+^ channels similarly to ventricle cardiomyocytes of neonatal rat.

## 3. Material and Methods

### 3.1. Animals

All experiments on animals were conducted according to the institutional animal care and use committee-approved protocol at Nagoya University (Aichi, Japan) and Nagaoka University of Technology (Niigata, Japan).

### 3.2. Id2 Transgenic Mice

Rat Id2 cDNA was isolated by RT-PCR with the forward primer obtained HindIII enzyme site 5′-GACAAGCTTATGAAAGCCTTCAGT-3′ and the reverse primer obtained EcoRV enzyme site 5′-ATAGATATCTTAGCCACAGAGTAC-3′ from rat heart total RNA. Rat Id2 cDNA was cloned into the plasmid vector containing α-myosin heavy chain (α-MHC) promoter which is heart specific isoform of MHC. This plasmid vector was kindly provided by Professor K. Otsu (King’s College London, London, UK). The α-MHC promoter-rat Id2-SV40 region was linearized and gel purified. The NPO for Biotechnology Research and Development performed microinjection into the pronuclear of DBF1 mouse zygotes. Transgenic (TG) mice were identified by PCR with the forward primer 5′-GCCCACACCAGAAATGAC-3′ that is specific sequence for α-MHC promoter and the reverse primer 5′-ATGCTGATGTCCGTGTTCAG-3′ that is specific sequence for Id2. Id2 TG mice were crossed with B6D2F1/Slc mice (SLC Japan, Shizuoka, Japan), and the male hetero F2 mice were used. The littermates of TG mice that did not contain transgene were used as control mice. All mice were analyzed at 8- to 12-weeks-old.

### 3.3. Aldosterone Treatment

The three types of mixed anesthetic agents (0.3 mg/kg of medetomidine, 4.0 mg/kg of midazolam, and 5.0 mg/kg of butorphanol) were used as anesthesia for experimental mice. The 8- to 12-week-old mice were implanted micro-osmotic pumps (Alzet, DURECT Co., Cupertino, CA, USA, model; 1007D) filled with 60 mg/Kg/day aldosterone (Sigma–Aldrich, St. Louis, MO, USA) or with saline subcutaneously. A week after the operation, animals were sacrificed by cervical dislocation. The hearts, lungs, and livers were weighed. Subsequently, the left tibia was dissected and the length of the bone was measured.

### 3.4. Cardiomyocyte Isolation and Culture

Isolation of neonatal rat ventricular myocytes were previously described [16]. The cardiomyocytes were transferred to culture plates, and fibroblasts were removed by adhesion onto the plates for 1 h. The cardiomyocytes were then plated overnight and maintained in DMEM (Nacalai Tesque, Kyoto, Japan) containing 10% fetal bovine serum (Gibco, Auckland, New Zealand), 2 mmol/L L-glutamine, 100 U/mL penicillin and 100 mg/mL streptomycin (Nacalai Tesque, Kyoto, Japan).

### 3.5. RT-PCR Analysis

Rat brain, atria, septum, right and left ventricle, end cardial, middle layer, and epi cardial were extracted from adult rat. The first-strand cDNA was synthesized using a commercial kit (ReverTra Ace qPCR RT Kit, Toyobo, Tokyo, Japan) according to the manufacturer’s instructions. Standard PCR was performed in a final volume of 25 μL containing synthesized cDNA (25 ng of RNA content), 400 nmol/L primers and commercial master mix (EmeraldAmp MAX PCR Master Mix, Takara, Otsu, Shiga, Japan). PCR conditions were 98 °C for 5 s, 60 °C for 5 s, and 72 °C for 60 s for 25–27 cycles with the final extension at 72 °C for 4 min. The PCR products were resolved by electrophoresis on a 2% agarose gel and stained with 0.5 μg/mL ethidium bromide. The whole pictures of RT-PCR experiments results shown in Appendix A are shown in Appendix A.

### 3.6. Real-Time Quantitative PCR Analysis

Real-time PCR was done with synthesized cDNA (50 ng of RNA content), 200 nM primers and commercial master mix (Power SYBR Green PCR Master Mix, Applied Biosystems, Warrington, UK) in a final volume of 25 µL and normalized to GAPDH. The PCR reaction was carried out in 96-well plates using Stratagene Mx3000P (Agilent Technologies). The primers used in the experiments are shown in Table 4.

### 3.7. Luciferase Reporter Assay

The rat *cacna1g* promoter region (−798/+694 bp) was generated by PCR with primers (5′-CTGTTTCACTCAGCAATGATC-3′ and 5′-GGTAGAAGTTGAGCACACAACGG-3′) using genomic DNA from neonatal rat ventricular myocytes. The *cacna1g* promoter region PCR fragment was subcloned into pUC118 vector using a commercial kit (Mighty Cloning Reagent Set (Blunt End), TaKaRa, Shiga, Japan). After confirming the direction of the fragment, approximately 1.5 kb of a KpnI/HindIII digested fragment was subcloned into the pGL3-basic vector (Promega, Madison, WI, USA).

On the second day after neonatal rat ventricular myocytes isolation, cells were infected by Id2-adenovirus. After 24 h, cells were transfected with 800 ng of the *cacna1g* promoter luciferase and 200 ng control vector pRL-null using Lipofectamine 3000 in Opti-MEM (Invitrogen, Carlsbad, CA, USA). Then, 24 h after transfection, cells were stimulated by 1 µmol/L aldosterone. After 24 h stimulation, luciferase activities were measured using dual-luciferase reporter assay system (Promega, Madison, WI, USA).

### 3.8. RNA Interference

Small interfering RNAs (siRNAs) targeting rat Id2 were designed with forward sequences (upper strand) as follows: Id2-2, AAACAGCCUGUCGGACCACATT and Id2-5, CCCGAUGAGUCUGCUCUACATT. These sequences were blasted against the NCBI database and showed no recognition of other rat ORFs. siRNAs were produced by (Operon, Tokyo, Japan) and transfected using a transfection reagent, Lipofectamine RNAiMAX Transfection Reagent, following the manufacturer’s instructions (Invitrogen).

### 3.9. Plasmid Construction

Rat Id2 cDNA was obtained by RT-PCR with the primers 5′-GCCTTTCCTCCTACGAGCAGCAT-3′ and 5′-AGCCACAGAGTACTTTGCT GTCATTC-3′ using total RNA isolated from the ventricular heart of adult male Sprague-Dawley rats. The obtained PCR product was cloned into pCR-BluntII-TOPO vector (Invitrogen, Carlsbad, CA, USA). Rat Id2 cDNA was inserted into the HindIII/SalI sites of the shuttle vector pDC316 for adenovirus production.

### 3.10. Adenovirus

Recombinant human adenoviruses-5 for gene delivery of Id2 was produced in HEK293 cells, amplified and purified by CsCl gradients as previously described [28].

### 3.11. Western Blot

Proteins from neonatal rat ventricle cardiomyocytes were extracted using a lysis buffer (in mmol/L: 50 Tris at pH 7.6, 400 NaCl, 5 EDTA, 1 DTT, 1 PMSF, 1% NP-40, 10% glycerol and a tablet of protease inhibitors; Roche). Proteins from freshly isolated mouse ventricle of control and Id2 transgenic mice were lysed in 200 µL lysis buffer. The obtained homogenates were centrifuged at 10,000 rpm for 5 min to remove any remaining large cellular fragments. Total cell lysates (60 µg proteins for calcium channels and 100 µg proteins for Id2) were resolved by 8% SDS–polyacrylamide gel electrophoresis (SDS–PAGE) for calcium channels and 15% gel for GAPDH and Id2. Proteins were transferred to polyvinylidene difluoride membranes (Millipore, Billerica, MA, USA) and immunoblotted with an anti-Id2 monoclonal antibody (Cell Signaling Technology, Danvers, MA, USA), an anti-CaV3.1 polyclonal antibody (Alomone Labs, Jerusalem, Israel), an anti-CaV1.2 polyclonal antibody (Alomone Labs, Jerusalem, Israel), anti-Cav3.2 polyclonal antibody (Alomone Labs, Jerusalem, Israel), an anti-GAPDH polyclonal antibody (Novus Biologicals, Littleton, Colorado, USA), and an anti-β-tubulin monoclonal antibody (Sigma–Aldrich, St. Louis, MO, USA). As secondary antibodies for detection, anti-mouse or an anti-rabbit IgG linked to horseradish peroxidase (Jackson Immuno Research, West Grove, PA, USA) were used, respectively. The whole pictures of western blots experiments results shown in Figure 1, Figure 2, Figure 6, Appendix A are shown in Appendix A.

### 3.12. Electrophysiology

Voltage-gated Ca^2+^ currents and spontaneous action potentials were recorded by the patch-clamp technique, as previously described [10]. Briefly cardiomyocytes were cultured on poly-lysine coated glass dishes. Recordings were performed in the whole-cell configuration using an Axopatch 200B Amplifier (Axon CNS, Molecular Devices, Sunnyvale, CA, USA) at a holding potential of −90 mV. Cells were depolarized from the −90 mV holding potential to −80 mV with +10 mV voltage steps of 200 ms. The current amplitude was measured at the peak. To establish the current density-voltage curve, the current density was determined with the cell capacitance. The mean ± SD cell capacitances were: control cells: 13.8 ± 2.73 pF; aldosterone treated cells: 20.3 ± 4.5 pF; Id2-Ad: 14.7 ± 5.1pF; Id2-Ad + aldosterone treated: 17.1 ± 4.2 pF; Id2-siRNA: 16.9 ± 4.46 pF; siRNA-luc: 15.4 ± 4.04 (*n* = 7–12 cells, as shown in Figure 4). The currents were filtered at 2 kHz and sampled at 5 kHz using a A/D converter, the Digidata 1440A (Axon CNS, Molecular Devices, Sunnyvale, CA). The leak was subtracted automatically by a P/4 protocol (pclamp10, Axon CNS, Molecular Devices, Sunnyvale, CA). For the recording Ca^2+^ currents, the bath solution contained (in mmol/L) 125 N-methyl-glucamine, 54-aminopyridine, 20 tetraethyl-ammonium chloride, 2 CaCl_2_, 2 MgCl_2_, and 10 D-glucose and was buffered to pH 7.4 with 10 HEPES. The patch pipettes were filled with solution containing (in mmol/L) 130 CsCl, 10 EGTA, 3 Mg-ATP and 0.4 Li-GTP with the pH adjusted to 7.2 by 25 HEPES. Spontaneous action potentials were recorded in the I = 0 mode of the Axopatch 200B Amplifier with a bath solution (in mmol/L): 146.9 NaCl, 5.4 KCl, 1.8 CaCl_2_, 0.33 NaH_2_PO_4_ and 5 Hepes, with the pH adjusted to 7.4 by NaOH and a pipette solution (in mmol/L): 60 KOH, 60 KCL, 40 Aspartate, 10 EGTA, 5 Hepes, 5 Mg-ATP, 5 Na-Phosphocreatine and 0.65 CaCl_2_ with the pH adjusted to 7.2 by KOH.

### 3.13. Statistical Analysis

Results are shown as the mean ± S.E.M. Statistical analysis was performed by a Student’s *t*-test for paired data or one-way ANOVA, followed by Tukey–Kramer’s post hoc test was used for multiple comparisons. ** p* < 0.05, ** *p* < 0.01, and *** *p* < 0.001.

## 4. Discussion

In the present study, we found that the transcriptional repressor Id2 expression is repressed by aldosterone in neonatal rat cardiomyocytes and the adult mouse heart. The overexpression of Id2 in cardiomyocytes prevented the aldosterone-stimulated an increase in spontaneous action potential rate while the knock down of Id2′s expression increased this rate similar with the aldosterone effect. Similarly, these effects were also observed for the expression of voltage-gated Ca^2+^ channels and Ca^2+^ currents. Taken altogether these results suggest that Id2 can function as a repressor of voltage-gated Ca^2+^ channels expression particularly CaV3.1 T-type Ca^2+^ channels.

The increase in voltage-gated Ca^2+^ channels expression (CaV1.2, CaV3.1, and CaV3.2) and their Ca^2+^ currents in cardiomyocytes stimulated with aldosterone was previously reported [11,12,26] and is consistent with our present results. Interestingly, when Id2 was overexpressed in neonatal rat ventricular cardiomyocytes, the aldosterone-stimulated increase in voltage-gated Ca^2+^ channels expression was suppressed, as well as the rate of their spontaneous action potentials, confirming their role in pacing the cardiomyocytes electrical activity. We confirmed in vivo the repressive effect of Id2 on voltage-gated Ca^2+^ channels expression in the heart of mice. These results thus confirmed that Id2 can block the aldosterone-stimulated transcription in cardiomyocytes. The interaction of Id2 with transcription factors prevents their binding to the promotor of their target genes preventing their expression [17]. The transcription factors whose activity is repressed by Id2 before its downregulation by aldosterone stimulation remain to be identify.

Aldosterone has been shown to up regulate or down regulate different type of genes, including transcription factors or transcriptional regulator [27]. Our results suggest that Id2 is another transcriptional regulator whose expression is regulated by aldosterone in the heart. The aldosterone stimulated repression is dependent on MR activation but does not appear to involve in the 3.1 kb promoter region of Id2 despite the presence of 21 predicted MR cis-binding consensus sequence. Therefore, the repression of Id2 might involve other regulatory mechanisms such as specific micro RNAs. The expression of the transcriptional repressor NRSF/REST is also down regulated by aldosterone and miRNA204, a miRNA upregulated by aldosterone, overexpression [16]. A similar mechanism involving a yet unidentified miRNA might downregulate Id2 upon aldosterone.

Interestingly, the siRNA knockdown of Id2 resulted only in an increase expression of CaV3.1 but not of CaV1.2 and CaV3.2 (Figure 3B). The absence of effect by the Id2 siRNA on CaV3.2 isoforms might be due to another transcriptional repressor, REST/NRSF, which represses also CaV3.2 expression but not CaV3.1 expression [29]. Thus, the repression of both Id2 and REST/NRSF should be necessary to enable the transcriptional activation of CaV3.2. Transcriptional repressors have been shown to provide cardiomyocytes a protective mechanism to counter the activation of pathological transcription [30]. The re-expression of T-type Ca^2+^ channels in the heart ventricle plays a role in the development of cardiac hypertrophy and heart failure [16]. The transcriptional repression of T-type Ca^2+^ could be therefore an efficient mechanism to protect the heart. Aldosterone excess would overcome the Id2 repression to stimulate the T-type Ca^2+^ increased re-expression. Further experience would be required to confirm this hypothesis and identify other potential targets of Id2 transcriptional repression.

In conclusion, taken all together our results suggest a pathway in which MR activation aldosterone stimulated the repression of Id2 expression, which in-turn might enable the re-expression of CaV3.1 T-type Ca^2+^ channels in ventricular cardiomyocytes triggering increased rate of spontaneous action potentials. In addition, the capacity of Id2 to prevent the aldosterone-stimulated increased expression of L- and T-type voltage-gated Ca^2+^ channels suggests a protective capacity for Id2 in cardiomyocytes. Further analysis in vivo would be necessary to confirm this hypothesis.

## Figures and Tables

**Figure 1 ijms-22-03561-f001:**
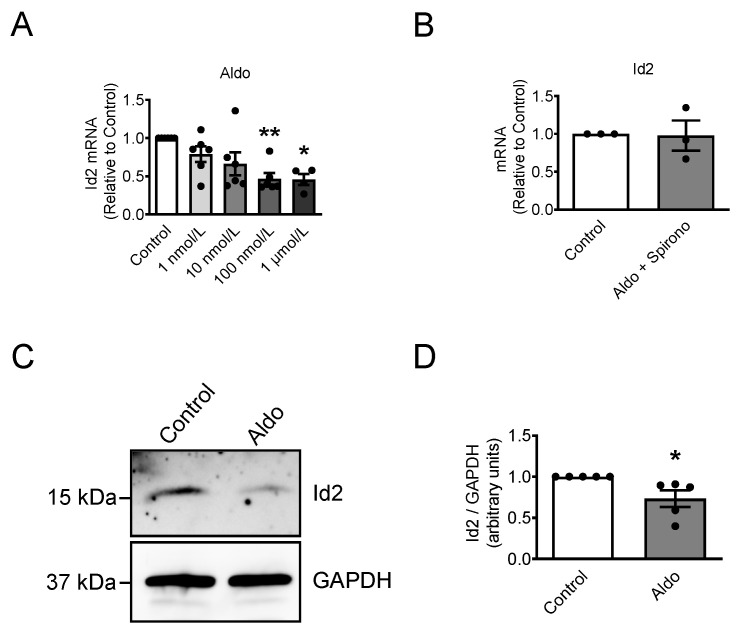
Aldosterone reduce the Id2 expression in neonatal rat ventricular cardiomyocytes. Neonatal rat cardiomyocytes were stimulated with (**A**) Aldo (aldosterone) or (**B**) 1 µmol/L Aldo + Spirono (Spironolactone) for 24 h. Bar graphs show the Id2′s mRNA level in aldosterone (Aldo)-treated and nontreated cardiomyocytes. Id2 mRNAs were measured by real-time qPCR analysis (*n* = 6 for (**A**) and *n* = 3 for (**B**)). (**C**) Representative pictures of western blot experiments showing the Id2 protein (upper) and Glyceraldehyde 3-phosphate dehydrogenase (GAPDH) protein (lower) expression levels in neonatal rat ventricular cardiomyocytes treated with 1 µmol/L aldosterone (Aldo) for 24 h. (**D**) The bar graph indicates the mean +/− s.e.m. of Id2 protein expression in aldosterone (Aldo)-treated (1 µmol/L) and non-treated cardiomyocytes from five independent experiments. The data were evaluated by one-way analysis of variance (ANOVA) followed by Tukey–Kramer’s post hoc test or by unpaired two-tailed Student’s *t*-test. Bar graphs are mean and + s.e.m, * *p* < 0.05, ** *p* < 0.01.

**Figure 2 ijms-22-03561-f002:**
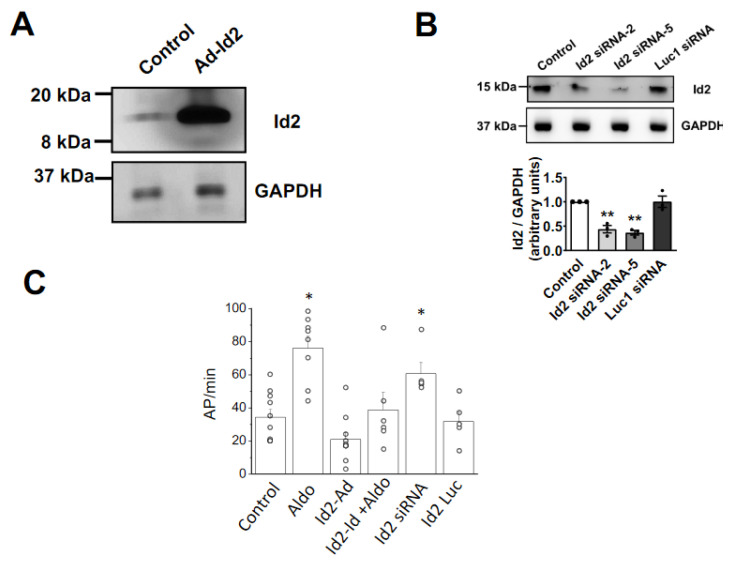
Effects of Id2 overexpression and siRNA knockdown on cardiomyocytes spontaneous action potential rate. (**A**) Id2 overexpression was achieved after 24 h transduction with adenoviral vector. Pictures of western blot experiments showing the Id2 protein (upper) and GAPDH protein (lower) expression levels in neonatal rat ventricular cardiomyocytes. (**B**) Id2 expression was knock down with siRNA. siRNAs were transfected for 48 h and proteins were extracted. Representative pictures of western blot experiments showing the Id2 protein (upper) and GAPDH protein (lower) expression levels in neonatal rat ventricular cardiomyocytes in control, Id2 targeting siRNA, and Luciferase (Luc) targeting siRNA. Bar graph are the mean of *n* = 3 independent experiments. (**C**) Bar graph shows the number of spontaneous action potential per min in control, aldosterone treated (1 µmol/L), Id2 overexpressing, Id2 overexpressing with aldosterone (1 µmol/L), Id2 siRNA, and Luc1 siRNA cardiomyocytes. The data were evaluated by one-way analysis of variance (ANOVA) followed by Tukey–Kramer’s post hoc test. Bar graphs are mean and + s.e.m, * *p* < 0.05, ** *p* < 0.01

**Figure 3 ijms-22-03561-f003:**
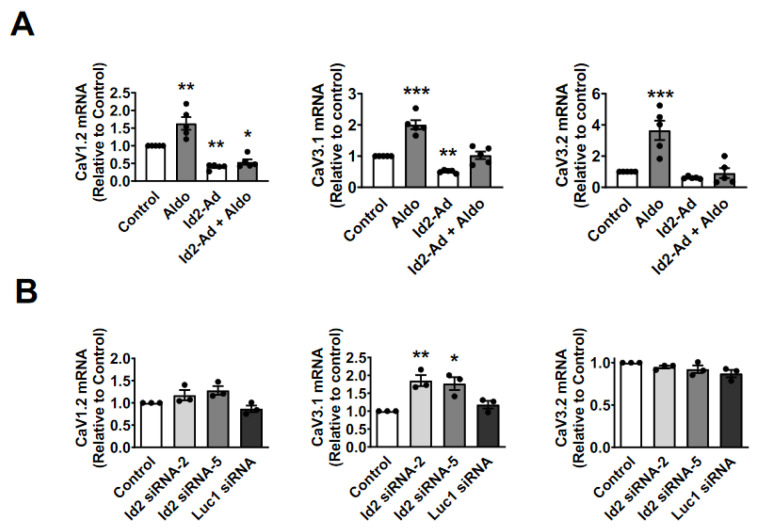
Manipulation of Id2′s expression altered of voltage-gated calcium channel expression in neonatal rat ventricular cardiomyocytes and prevented aldosterone-stimulated increased. (**A**) 2, CaV3.1, and CaV3.2 were measured by RT-qPCR in neonatal rat ventricular cardiomyocytes treated and non-treated with 1 µmol/L Aldo for 24 h upon Id2 overexpression. Bar graphs shows the mean expression of CaV1.2 mRNA (left), CaV3.1 mRNA (middle), and CaV3.2 (right) (*n* = 5). (**B**) The mRNA expression of CaV1.2, CaV3.1, and CaV3.2 were measured by real-time qPCR in neonatal rat ventricular cardiomyocytes treated with Id2 siRNA-2, siRNA-5 or Luciferase1 siRNA for 48 h. Bar graphs shows the mean expression of CaV1.2 mRNA (left), CaV3.1 mRNA (middle), and CaV3.2 (right) (*n* = 3). The data were evaluated by one-way analysis of variance (ANOVA) followed by Tukey–Kramer’s post hoc test. Data are the mean ± s.e.m, * *p* < 0.05, ** *p* < 0.01. *** *p* < 0.001.

**Figure 4 ijms-22-03561-f004:**
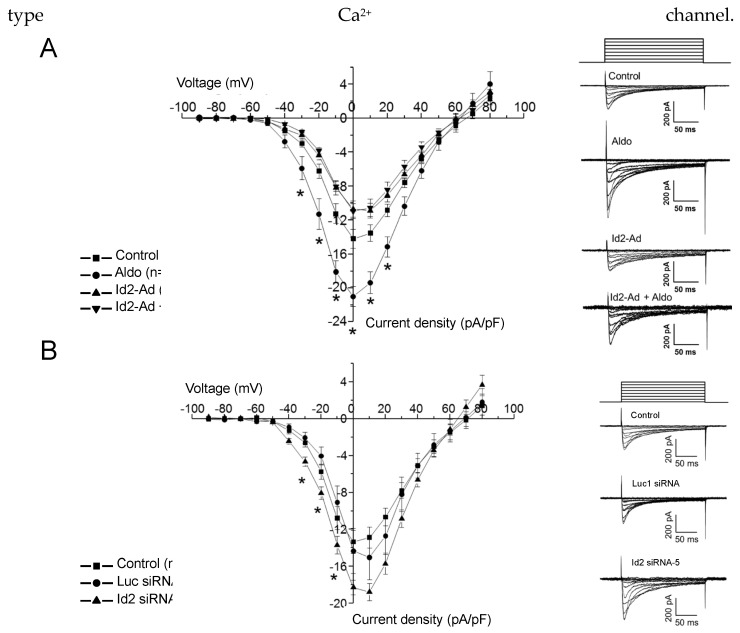
Manipulation of Id2′s expression altered of voltage-gated Ca^2+^ currents in neonatal rat ventricular cardiomyocytes. Voltage-gated Ca^2+^ currents were measured by whole-cell patch-clamp experiments under the voltage-clamp mode. Graphs show the mean current-voltage relationship. currents measured in ventricular neonatal rat cardiomyocytes. (**A**) Graphs show the mean current-voltage relationship (IV curve) of voltage-gated Ca^2+^ currents measured in ventricular neonatal rat cardiomyocytes. Ca^2+^ currents were measured from a holding potential of −90 mV and depolarized by 10 mV voltage steps to +80 mV, as described in Methods. (A) IV curves recorded from cells that were treated (Aldo, ●☐) or non-treated (Control, ■) with 1 µmol/L aldosterone for 24 h and in the same cells overexpressing Id2, treated (Id2-Ad + Aldo, ▲) or non-treated (Id2-Ad, ▼) with aldosterone (Aldo) at 1 µmol/L. Right traces are representative recordings traces for cells non-treated (Control), treated (Aldo) with aldosterone at 1 µmol/L (Aldo) (n=10 to 11 cells). (**B**) IV curves of Ca^2+^ currents recorded in non-treated cardiomyocytes (■) and cardiomyocytes transfected with Id2 siRNA (siRNA Id2-5, ▲) or control siRNA (siRNA Luc1, ●☐) (*n* = 7–12 cells). Data are the mean ± s.e.m of 7 to 12 individual cardiomyocytes, as indicated in the figure for each condition from 3 to 4 different cell isolations. * *p* < 0.05, from unpaired student’s *t*-test.

**Figure 5 ijms-22-03561-f005:**
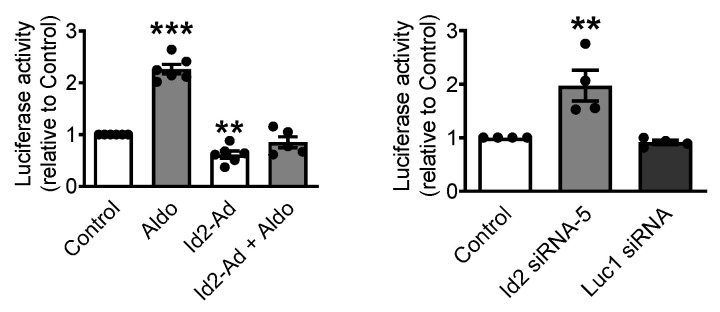
Id2 regulates the activity of CaV3.1 promoter. The luciferase reporter gene was placed under the control of 0.8kb of the CaV3.1 promoter. The reporter gene was transected into neonatal rat cardiomyocytes to measure the luciferase activity upon Id2 overexpression (left) or Id2 siRNA knockdown (right). Bar graphs represent the mean of the relative activity of luciferase from the CaV3.1 luciferase reporter plasmid construct. The data were evaluated by one-way analysis of variance (ANOVA) followed by Tukey–Kramer’s post hoc test. Values are mean + s.e.m., *** *p* < 0.001, ** *p* < 0.01 (*n* = 4–6).

**Figure 6 ijms-22-03561-f006:**
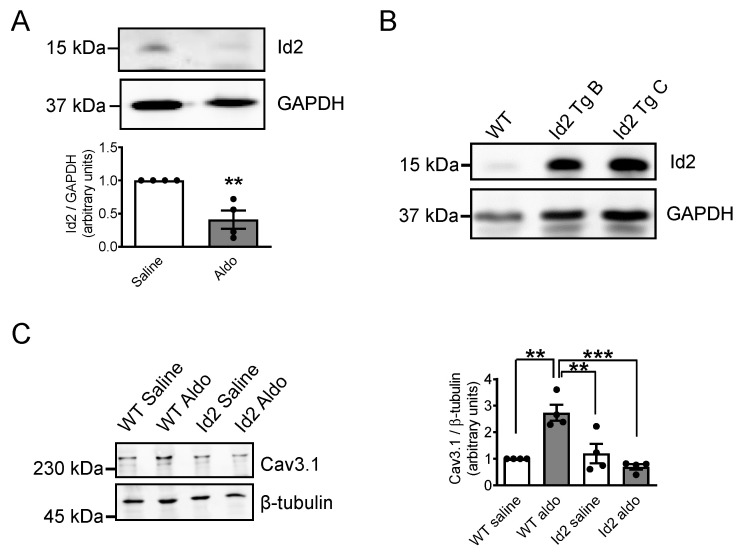
Id2 expressing transgenic mice prevents the aldosterone-stimulated expression of CaV3.1 voltage-gated calcium channels in vivo. WT or cardiomyocyte-specific Id2 expressing transgenic B6D2F1/Slc mice were stimulated with Aldosterone (60 mg/Kg/day) or Saline solution for 1 week by implanting subcutaneously osmotic pumps. (**A**) Representative pictures of western blot experiments showing the Id2 protein (upper) and GAPDH protein (lower) expression levels from the heart of WT mice treated with saline solution or aldosterone. The bar graph is the mean of Id2 expression (*n* = 4). (**B**) Pictures of western blotting experiments showing the Id2 protein (upper) and GAPDH protein (lower) expression levels in the heart of B6D2F1/Slc WT and two cardiomyocyte-specific Id2 expressing transgenic (Tg) mice. (**C**) Pictures of western blotting experiments showing CaV3.1 (upper) and tubulin (lower) proteins expression levels in WT or Id2 transgenic mice treated with saline solution or aldosterone. Graph is are the mean expression of CaV3.1 and tubulin expression (*n* = 4). The data were evaluated by one-way analysis of variance (ANOVA) followed by Tukey–Kramer’s post hoc test or by unpaired two-tailed Student’s *t*-test. Bars and error bars indicate the mean ± s.e.m., ** *p* < 0.01, *** *p* < 0.001.

**Table 1 ijms-22-03561-t001:** Physiological characteristics in transgenic mice at baseline.

	WT (*n* = 5)	Id2 Tg B (*n* = 5)	Id2 Tg C (*n* = 5)
Body weight (g)	21.8 ± 1.8	21.1 ± 1.3	21.0 ± 0.5
Whole heart weight (mg)	120.9 ± 8.4	119.2 ± 8.5	117.4 ± 7.6
Lung weight (mg)	109.1 ± 4.3	108.8 ± 2.9	111.6 ± 2.8
Liver weight (mg)	1004.2 ± 13.6	973.3 ± 69.8	1014.2 ± 37.3
WHW/BW	5.54 ± 0.39	5.64 ± 0.15	5.58 ± 0.29

**Table 2 ijms-22-03561-t002:** Physiological characteristics of mice 1 week after saline and aldosterone infusion.

		WT Saline (*n* = 8)	Id2 Saline (*n* = 6)	WT Aldo (*n* = 8)	Id2 Aldo (*n* = 5)
Body weight (g)		24.3 ± 1.2	23.8 ± 0.8	25.1 ± 0.4	24.3 ± 1.1
Whole heart weight (mg)		127.5 ± 13.2	119.1 ± 9.6	155.8 ± 15.8	133.9 ± 5.8
Lung weight (mg)		121.3 ± 10.3	111.9 ± 3.6	137.9 ± 9.6	126.7 ± 4.0
Liver weight (mg)		1129.9 ± 77.4	1076.3 ± 63.3	1105.6 ± 37.9	1112.1 ± 45.3
WHW/BW		5.22 ± 0.43	4.97 ± 0.28	6.20 ± 0.61	5.51 ± 0.11

**Table 3 ijms-22-03561-t003:** Characteristics of mice 1 week after saline and aldosterone infusion.

		Saline (*n* = 5)	Aldo (*n* = 8)	*p* Value
Body weight (g)		23.8 ± 1.2	25.2 ± 0.8	0.318
Whole heart weight (mg)		118.4 ± 14.2	149.8 ± 17.9	0.244
Lung weight (mg)		116.1 ± 9.2	140.0 ± 10.2	0.138
Liver weight (mg)		1081.6 ± 93.7	1141.6 ± 71.8	0.618
WHW/BW		4.95 ± 0.46	5.96 ± 0.70	0.319

**Table 4 ijms-22-03561-t004:** Primer sequences used for RT or real-time PCR.

	Forward 5’-3’	Reverse 5’-3’
Rat Id2	CTCCAAGCTCAAGGAACTGG	ATGCTGATGTCCGTGTTCAG
Rat CaV1.2	AGCAACTTCCCTCAGACGTTTG	GCTTCTCATGGGACGGTGAT
Rat CaV3.1	ACGCTGAGTCTCTCTGGTTTGTC	TGCTTACGTGGGACTTTTCAGA
Rat CaV3.2	GGCGAAGAAGGCAAAGATGA	GCGTGACACTGGGCATGTT
Rat GAPDH	CAACTCCCTCAAGATTGTCAGCAA	GGCATGGACTGTGGTCATGA
Rat Hcn2	GGACACTTTCTTCCTCATGGAC	CTCCGTGTTGTCCTCAATAAC
Rat Hcn4	CTTCCTCATTGACTTGGTCCTC	TACTTCATCTTGATCCGCTGTG
Rat Kcnh2	ACCTTCAACCTTCGAGATACCAAC	GCTCTGTGTCCTTGTCAGTACG
Rat Kcnd3	GACAAGAACAAGCGGCAAGA	GCATTCATAGCGTGGGTAGT
Rat Scn5a	ACCTGCCTCTGAACTACACCATC	CCTTGGTCCAGTACAACTCTCC

## Data Availability

Please refer to suggested Data Availability Statements in section “MDPI Research Data Policies” at https://www.mdpi.com/ethics (accessed on 30 March 2021).

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
