# Peer review of "Id2 Represses Aldosterone-Stimulated Cardiac T-Type Calcium Channels Expression"

_ijms, 2021, doi:10.3390/ijms22073561_

Round 1

Reviewer 1 Report

The manuscript “Id2 represses aldosterone-stimulated cardiac T-type calcium channels expression”, is a very a well performed study about the role of the role of the Inhibitor of differentiation/DNA binding protein 2 (Id2) and its function as a transcriptional repressor for L- and T-type Ca2+ channels, particularly in cardiomyocytes.

In my opinion, this is a well designed and performed study. The manuscript is written very well and I do not find any significant incorrectness. My following comments are of minor character:

- Line 17: The word “Abstract”, or then, “Figure 1”… should not be underlined

- Line 90: Please define “MR” receptor

- Line 94: The font of Figure 1 should be reduced

- Line 104: “increase” should be “increased”

- Figure 2B is too small

- Line 162: The sentence “Aldosterone stimulation with 1 umol/L stimulation” must be rephrased

- Figure 4 and Figure 5: see Figure 1

- Please check and correct the reference list

Author Response

Reply to Reviewer 1 are on the attached word file.

Reviewer 2 Report

The subject of the article is very important, once aldosterone is associated with the development of heart pathologies, in particular hypertrophy, arrhythmia and heart failure. Moreover, the authors suggest that Id2 can function as a repressor of voltage-gated Ca mainly CaV3.1. The article is also very complete, since many complementary techniques are presented.

General comments:

  • The objective of the work must be clearly defined, in the last paragraph of the introduction.
  • The results cannot be included in the introduction, please rephrase the last sentence of the introduction.
  • In the results section the bibliographic references should only be necessary. So the phrase “Id2 is interesting, because it is a transcriptional repressor important for the development of heart conduction system during embryogenesis [18, 25]” must be should be placed in the introduction or discussion.
  • Introduced the statistic tests used in the legends of the figures.
  • In figure 1 the D is missing and a description of the aldo concentration is missing. It also doesn't have the various abbreviations.
  • In Fig. 4 B, is there no lack of aldosterone currents?
  • In my opinion the protocol that was used in the pacth clamp should be placed. The number of cells where the activity was analyzed must be added.
  • What is the average cell size (PF)?
  • The discussion is very small, and in my opinion is only a conclusion. Part of the bibliographic references from the results section should come for the discussion.

Author Response

Reply to Reviewer 2 are on the attached word file.

Round 2

Reviewer 2 Report

The authors have revised the manuscript according to the reviewer's comments.

I recommend accept